# Impressive Nasal Septum Regeneration after Cord Blood Platelet Gel (CBPG) in Extreme Premature Neonate with Non-Invasive Ventilation: A Case Report

**DOI:** 10.3390/children9111767

**Published:** 2022-11-17

**Authors:** Raffaele Falsaperla, Giulia Marialidia Biondi, Milena Motta, Pasquale Gallerano, Giusi Tancredi, Piero Pavone, Martino Ruggieri

**Affiliations:** 1Neonatal Intensive Care Unit, AUO Policlinico “Rodolico-San Marco”, University of Catania, 95121 Catania, Italy; 2Postgraduate Training Program in Pediatrics, Department of Clinical and Experimental Medicine, University of Catania, 95123 Catania, Italy; 3Transfusional Medicine, Complex Operative Unit, PO “Giovanni Paolo II”, 92019 Sciacca, Italy; 4Section of Pediatrics and Child Neuropsichiatry, Department of Child and Experimental Medicine, University of Catania, AOU “Policlinico PO” G. Rodolico, 95123 Catania, Italy; 5Unit of Rare Diseases of the Nervous System in Childhood, Department of Clinical and Experimental Medicine, Section of Pediatrics and Child Neuropsichiatry, University of Catania, AOU “Policlinico PO” G. Rodolico, 95123 Catania, Italy

**Keywords:** platelet gel, nasal septum, septal tissue, prematurity, non-invasive ventilation

## Abstract

Background: We evaluated the efficacy of Cord Blood Platelet Gel (CBPG) in the regenerative reconstruction of the nasal septal tissue of a preterm infant undergoing non-invasive ventilation. Methods: A CBPC treatment was used to enhance the regeneration of the nasal septum of a premature patient in an experimental way, evaluating the efficacy described in the literature (selective bibliographic search in PubMed) of the use of blood products for non-transfusion purposes. Results: A partial but satisfactory regeneration of the patient’s nasal septum was observed. Using the free NIH Image J online software, we were able to calculate the regenerated surface (about 83% of the destroyed cartilage). Conclusions: The use of platelet gel has been a promising alternative to surgical treatment in patients with severe damage to the nasal septum.

## 1. Introduction

Nasal trauma is considered to be one of the most serious and frequent complications of nasal tips Continuous Positive Pressure Ventilation (CPAP) use [1,2,3,4,5,6]. Binasal tips are the most commonly used interface for the delivery of positive nasal airway pressure (CPAP) in preterm infants [7,8]. Due to pulmonary immaturity, surfactant deficiency, and weak respiratory drive, up to 82% of preterm infants still require non-invasive ventilation post extubation to maximize oxygenation and ventilation [7]. Force applied to the tissues of the nostrils and nasal septum of the most exposed preterm infants can compromise skin integrity and cause severe nasal injury [9]. The onset of nasal injuries is multifactorial and is in relation to the duration of non-invasive ventilation in the premature subject, which is more susceptible to infections, and his integumentary system is more fragile and vulnerable [4,5,6].

Five categories of primary causes of nose injuries can be distinguished: material hardness, health care, neonatal clinical conditions, professional competence, and equipment. The interface pattern and the sterilization process are associated with material hardness [6].

In the literature, the incidence of nasal injuries ranges from 15–20% to 60% in the neonatal population undergoing this mode of neonatal ventilation, and there are high rates if a younger gestational age is present [5].

A Brazilian prospective observational study including 28 preterms evaluated the severity and incidence of internal and external nasal lesions after short nasal tip ventilation for more than 24 h: the incidence of internal injuries was 71.43%, and the incidence of external nasal lesions was 67.86% [6]. Bonfim et al. [10] proved that the incidence of nasal septum injuries found in infants corresponded to 62.9%.

Preventive maneuvers to avoid nasal lesions include interface modification [11]. Occasionally, if the damage is severe, surgical repair may be necessary [5].

Scientific evidence suggests the clinical use of the non-stem components of cord blood (platelets, plasma, and red blood cells). In particular, plasma can be exploited as eye drops, while platelet gel and lysate can be useful in the treatment of skin ulcers as well as eye drops. The red blood cells obtained at present have only been used experimentally in neonatal transfusions [12]. The procedural model of platelet gel collection used is that of Rebulla et al. [13].

The platelet gel used is a blood component for non-transfusion use rich in growth factors that allow the acceleration of tissue regenerative processes, and umbilical cord blood is richer in growth factors than adult blood. The model involves collection of the cord in the delivery room and subsequent transfer, with informed consent, to the transfusion center for hospital use or to the cord bank for transplantation/inventory [14,15].

Recently, platelet gel application was performed on two infants who had occipital pressure ulcers, a rare occurrence in the neonatal population. Cord Blood Platelet Gel (CBPG) application on occipital pressure ulcers was performed every 48 h. In both cases, a fast improvement was obtained since the first application. Healing occurred after a total of 8 applications in one case and after a total of 14 applications in the other [14].

Aiming to improve nasal septum impairment in the department of Neonatology, we carried out an experimental topical application of CBPG obtained from umbilical cord blood to an extremely premature infant who had suffered severe nasal septum destruction from the CPAP interface.

Recent reports described in the literature for uses of the CBPG topically [14,16,17,18,19] prompted us to try to use it on the damaged nasal structure of an infant when he was about 8 weeks old.


**Case Report**


This 12-month-old male was born at the 25th week of gestation with weight of 730 g, height of 46 cm, and occipito-frontal circumference of 33 via an emergency cesarian section for maternal respiratory complications from SARS-CoV-2 infection.

At birth, the baby was intubated for one and a half month. 

Afterward, he required minimal ventilation for another month through the use of nasal cannulae ventilation. The prolonged use of CPAP and the fragility of the structures of the preterm caused the destruction of the nasal septum in the baby one month after the extubation. The first lesion of the nose started as simple hyperemia and then progressed to a major septal lesion two week after the extubation.

Various preventive measures, such as cortisone cream and hyaluronic acid cream, were applied to no avail.

## 2. Materials and Methods

Periodic applications were carried out about 1–2 a day, with changes seen even after a few days and with vast improvement after 5–6 months. We used Fischer’s classification [4] for our patient’s nasal trauma. According to the classification proposed by Fischer et al. [4]. Stage I, intact skin with non-blanchable erythema; Stage II, partial loss of dermis thickness, presenting as a superficial wound, red bed, no crust; and Stage III, necrosis and total tissue loss [4].

Pre-treatment stage was type III (Figure 1), As reported by Ribeiro et al. [6], external nasal lesions were classified as Fischer’s Stage I (68.42%) and Stage II (31.58%). All internal injuries were evaluated as Stage II. The medical staff was trained to carefully observe the nose externally every 6 h during nCPAP treatment, with the removal of the interface to allow closer local inspection. CBPG was applied 1 mL twice daily for a total of 15 days. The CBPG was obtained by adding thrombin, batroxobin, and/or calcium to a platelet concentrate. The preparation of the CBPG was carried out using the BioNest system, described by Rebulla et al. [12,13]. The BioNest system is a bag with an easy opening for the extraction and therapeutic application of CBPG. To obtain it, extraction is carried out, producing the platelet concentrate, which is frozen (−30 °C) and subsequently activated with the addition of calcium gluconate. The platelet concentration is approximately 800 × 109 platelets/L + 20%. The product must be used within 90 min of activation. 

## 3. Results

As reported in the figures with reconstruction and from the graphic, during the treatment in the patient, there was a gradual transition from the lack of the nasal septum to gradual regeneration. Moving from stage III to stage II represented by the presence of superficial erosion to a partial but evident regeneration after a few weeks (Figure 1). The quantitative analysis was performed using the free online software NIH ImageJ. We drew the ROI (region of interest) of the area that we wanted to measure using the function freehand selections, corresponding to the red area of the figures, in which the orange area corresponds to the platelet gel application and the black area corresponds to the nostrils (Figure 1). We then used the graphical measurement function to calculate the value of these red areas, as shown in Figure 1. Included there is the photo of the nasal septum of a child of the same age in normal conditions and on the basis of the measured area we calculated the percentage that in this clinical case is reconstructed (Figure 1). The figure shows the timeline from the initial condition of the patient before the platelet gel application to the final recovery with partial but satisfactory reconstruction of the nasal septal cartilage of the nasal septum (approximately 83%). The reconstructed area is portrayed in red in Figure 1.

## 4. Discussion

The infant showed a severely damaged nasal septum, and various preventive measures had failed. On the basis of the result of the literature, we tried a tentative solution. We found the results obtained satisfactory. 

### 4.1. How CBPG Is Obtained

CBPG is composed of plasma rich in platelets and red blood cells. The hemocomponents for non-transfusion use, which are the subject of this article, are those set out in the Ministerial Decree of 1 August 2019—Amendments to the Decree of 2 November 2015, on: ‘Provisions relating to the quality and safety requirements for blood and haemocomponents’, which distinguishes between products of platelet origin, products of plasma origin and products of serum origin. 

The use of blood components for non-transfusion use has become widespread in various specialist fields of medicine and surgery. In relation to the methods of use of blood products for non-transfusion use, these can be used for application to the skin or mucous surfaces, topic, intra-tissue or intra-articular infiltration, or locally in surgical sites.

In addition to their widely prevalent use for hemorrhage prophylaxis, platelet concentrates have also been used for many years to prepare platelet gels for topical skin ulcer repair. Platelet gel can be obtained by activation of fresh, cryopreserved, autologous, or allogeneic platelet concentrates with calcium gluconate, thrombin, and/or batroxobin. The high content of diffuse tissue regenerating factors in umbilical cord cells and the availability of allogeneic cord blood units donated for hematopoietic transplantation but unsuitable for this use due to the low content of hematopoietic stem cells have made it possible to hematopoietic stem cell content have enabled the development and standardization of cryopreserved allogeneic cord blood platelet concentrates (CBPCs) suitable for subsequent preparation of clinical-grade cord blood platelet gels that can be used for regenerative purposes.

The cord blood units used for the production of platelet gel come from voluntary donations made at the birth points of Sicilian health facilities connected to the Cord Blood Bank of Sciacca (Italy).

Cord blood units collected immediately after the birth of the baby come from couples who have signed an informed consent and have been assessed for the absence of exclusion criteria for donation (as provided for by current transfusion regulations for blood-borne diseases).

Cord blood units are collected using sterile collection devices in bags containing 21–29 mL of citrate-phosphate-dextrose (CPD) anticoagulant by trained midwives, either during physiological delivery or cesarean section, according to validated standard operating procedures distributed by the Cord Blood Bank of Sciacca belonging to the Italian Cord Blood Network coordinated by the National Blood Centre.

After storage and transport at a controlled temperature of <10 °C, the units are processed to prepare CBPG within 48 h of collection.

The unit undergoes double centrifugation to obtain a volume of 5–15 mL of platelet-rich plasma with a target mean platelet concentration of 1 × 109/L (range 0.8–1.2 × 109/L).

Initial centrifugation of the unit is at 210× *g* × 10 min, followed by transfer of the platelet-rich plasma (PRP) to a secondary bag and centrifugation of the PRP at 2100× *g* × 15 min.

At this point, removal of the supernatant platelet-poor plasma (PPP) in excess compared to the final target volume of CBPC and frozen at −30 °C.

Platelet gel preparation, at the clinician’s request, consists of thawing the platelet concentrate placed in bagsteryl, transferring it to a sterile qualified device, and adding calcium gluconate at a concentration of 1:2 to achieve gelation.

Within 90′ of gelation, the sample is transported to the site of clinical use and applied to the skin lesion.

### 4.2. How Does CBPG Act?

The reparative mechanism triggered by the application of platelet gel is scientifically based on the local application of growth-stimulating factors contained in the platelet granules, which trigger a signal transduction mechanism amplifying and accelerating the regenerative process.

By applying hyperconcentrated platelets activated with Calcium Gluconate in the form of platelet gel at the site of the injury, tissue regeneration processes are triggered and accelerated. 

Platelets can be compared to cellular reservoirs that process, store and then release, when activated, numerous growth factors capable of stimulating the proliferation of stem cells, mesenchymal cells, fibroblasts, osteoblasts, keratinocytes, melanocytes, pericytes, myocytes, chondrocytes, and endothelial cells, also exerting a chemotactic action on macrophages, monocytes, and polymorphonucleates. Platelet gel is a hemocomponent that, when used topically (topical hemotherapy), in an adjuvant sense, enhances the natural local processes of hemostasis, adhesion, and repair. The purpose of its use is to accelerate the mechanisms leading to tissue regeneration and wound healing. The literature is now very rich in reports concerning the topical use of GFs contained in α-granules (PDGF, TGF-β, EGF, FGF, VEGF, and IGF-1), and platelets constitute an inexhaustible source of clinically employable Growth Factors (GFs). Therefore, the ability of platelets to intervene in tissue regeneration mechanisms is the theoretical prerequisite for the production and clinical use of platelet gel [16,17,18,19,20]. 

Platelet gel using cord blood provides a rich and unique combination of nutritional factors, as well as high levels of molecules such as PDGF and FGF, which collectively suggests that platelet gel from cord blood promotes MSC proliferation in vitro in the absence of serum, particularly between days 1 and 8 compared to standard or osteogenic media Platelet-rich plasma gel also promotes over-regulation of alkaline phosphatase and increases mineral and type I collagen deposition during osteogenic differentiation of MSCs. 

### 4.3. CBPG Would Also Fall within the Scope of the Proposed Legislation on Substances of Human Origin (SoHO) of the European Commission Due to Be Finalized

Blood components for non-transfusion use, due to their proven regenerative, tissue repairing, and wound healing capabilities, are widely used in clinical settings belonging to different specialist branches of medicine and surgery in public and private health care regimes.

Autologous and allogeneic blood components for non-transfusion use are prepared within the Transfusion Services (TS) and their organizational articulations in accordance with the provisions of the transfusion regulations in force regarding the collection, preparation, biological qualification (where applicable), storage and distribution (Decree of 2 November 2015 and Ministerial Decree of 1 August 2019—Amendments to the Decree of 2 November 2015, bearing: ‘Provisions on quality and safety requirements for blood and blood components’) [5,16].

### 4.4. Clinical Usage of CBPG

According to the most recent literature, Cord Blood Plasma (CBP) has been shown to be richer in Tumor Growth Factor (TGF)-β2 and TGF-β3, Epidermal Growth Factor (EGF), Hepatocyte Growth Factor (HGF), Platelet-Derived Growth Factor (PDGF-BB) and Vascular Endothelial Growth Factor (VEGF)-A and D, and with a lower concentration of Insulin-Like Growth Factor (IGF-I) and IGF-II than adult plasma [15]. An analysis of CBP lysates with PG, performed by Longo et al. [21], revealed that the composition of the umbilical CB proteome consists of elements such as dermatopontin, procollagen C endopeptidase enhancer-1 and lumican directly involved in the process of re-epithelialization and wound healing.

CBPG has been shown to be a valuable treatment in numerous conditions such as diabetic foot ulcers, pressure ulcers, epidermolysis bullosa, oral mucositis, fistulae, and surgical wound dehiscence [22]. A prospective clinical trial, which enrolled 20 diabetic patients with lower limb ischemia, reported promising results after the application of platelet gel on post-revascularization foot ulcers [21]. Piccin et al. [23] demonstrated tissue regeneration in severe mucositis injury in patients with non-Hodgkin’s lymphoma after the application of cord platelet gel for 8 consecutive days without any undesirable effects. Bisceglia et al. [24] performed the first endocavitary treatment of a form of liquid platelet gel in a patient with recurrent perianal fistula with encouraging results. The use of CBPG was found to be very effective in the treatment of epidermolysis bullosa in three studies, one of which was postoperative [25,26,27]. In a pilot study that enrolled three children with epidermolysis bullosa, two showed a greater improvement in skin lesions treated with CBPG compared to lesions treated with standard drugs, one with a result comparable to standard therapies but with better skin quality [28].

## 5. Conclusions

The application of the platelet gel in the patient induced and facilitated the reconstitution of the septal cartilage tissue with excellent results from the first applications, but more surprising after several applications and several weeks. CBPG contains many angiogenetic and growth factors; these characteristics make it indicated in treating soft tissue injuries. It would seem to be a safe and effective treatment in neonates [28]. The use of platelet gel has been a promising alternative to surgical treatment in patients with severe damage to the nasal septum. The present report shows some limitations: (a) non-blinded, freehand application of the imaging analysis system; (b) no check-control; (c) no valid proposition should be used on a single case; (d) several high-quality clinical trials are necessary to reach valid results. The present experience could be considered one of the starting points for new clinical trials for the treatment of lesions with significant septal necrosis in premature infants undergoing non-invasive ventilation for long periods.

## Figures and Tables

**Figure 1 children-09-01767-f001:**
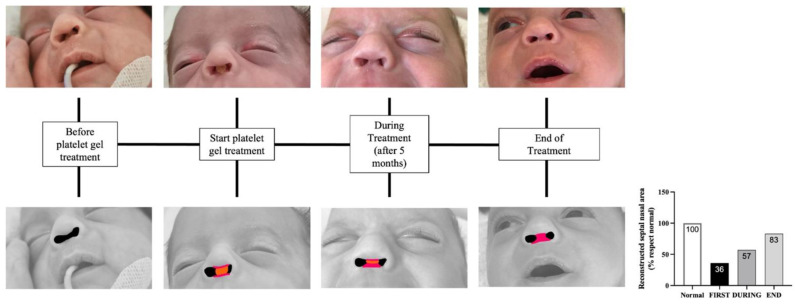
The reconstructed area is portrayed in red in the figure below.

## Data Availability

Not applicable.

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
