# Peer review of "Impressive Nasal Septum Regeneration after Cord Blood Platelet Gel (CBPG) in Extreme Premature Neonate with Non-Invasive Ventilation: A Case Report"

_children, 2022, doi:10.3390/children9111767_

Round 1
Reviewer 1 Report
Dear Editor, thank you for inviting me to review this interesting case presentation.
The authors shared their experience in using a cord platelet gel to ameliorate a nasal trauma induced by a nasal interface from non-invasive ventilation. In general, reporting the use of novel therapies in single patients is worthwhile for several reasons.
The main problem with this report, however, is that no one knows to what extent the experimental treatment can be made responsible for the partial recovery of the nasal septum in this patient.
However, to increase the value of this manuscript for the reader, following suggestions are made:
1. Please try to be concise: the introduction should start from line 42 (“nasal trauma …”). No key information or background needed to understand the content and message of the manuscript will be lost. Figure 1 may rather be omitted and the issue covered in a few sentences.
2. Please try to be clear: the introduction is very confuse und unstructured until line 63. Neither the sentences not the content appear to follow a logic order.
3. CBPG may be known to some, but most likely not all readers. Therefore, please explain in detail what constitutes this gel, how is it produced and where does the cord blood come into play? This does not become apparent to the reader.
4. What is the supposed mode of action of CBPG? Line 111: “The reparative mechanism is triggered by the application of the gel” is insufficient as an explanation! Move this from the methods section to the introduction as this is your intervention!
5. Case Report: Lines 79 to 86 are irrelevant and can be omitted. Please detail: What kind of nasal interfaces were in use? What is the local standard of care to avoid nasal injuries? Give exact times for invasive and non-invasive ventilation. At what day of age was the first injury detected and how did it progress. What measures were taken in that individual patient to improve the situation?
6. Material and Methods: Please do not state “surprising results” (line 94) in the methods section. Please try to quantify: how much gel was applied per treatment? Please correct “Fisher” in line 101.
7. Results: Please move the details of the image analysis to the methods section. Please inform the reader about the nature of the 3D reconstruction, as there are no actual 3D images depicted in the manuscript.
8. Discussion: Line 158-160 may be moved to the Introduction to support the rationale for using CBPG in the first place. Please add limitation concerning the non-blinded, freehand application of the imaging analysis system. Another limitation was mentioned before: there are no controls.
9. Conclusion: I would advise the authors to scale down their conclusion, as no valid proposition should be based on a case report but rather on high quality clinical trials.
10. As CBPG would also fall within the scope of the proposed legislation on substances of human origin (SoHO) of the European Commission due to be finalized: Please inform the reader about the current regulatory situation in your area of jurisdiction when dealing with CBPG and its manufacturing.
Reviewer 2 Report
I wish to thank the authors for a highly original and relevant case report. Some moderate English modifications are required (see following suggestions):
Abstract;
Lines 19-20 We thought of using a platelet gel..... suggest changing to... A platelet gel was used to enhance regeneration of the nasal septum....
Line 21: Suggest changing the word succes to efficacy
Line 22: Suggest removing We witnessed.. in the results section. Start sentence with.. A partial but satisfactory....
Line 23: Add "was observed" to the end of the sentence...regeneration of the nasal septum was observed
Introduction:
Line 56...preterm infants still require MV psot extubation... suggest changing MV to non invasive ventilation....
Line 56....post extubation to check the level... change to ..post extubation to maximize oxygenation and ventilation.
Line 73.. At the aim to improving... suggest changing to... Aiming to improve nasal septum....
Line 75-76.. who had suffered injuries from permanence of ventilatory support to totally destroy nasal septum... suggest changing to... who had suffered severe nasal septum destruction from the CPAP interface.
Line 79. suggest removing.... in the delivery room from.... suggest changing to... via an emergency caesarian section...
Line 82: He was then intubated.. remove the word then
Line 85: Clarification: Did the authors mean to write IV caffeine or did they mean IV epinephrine? Also add was given to the end of that sentence.
Line 87: .. he was given non invasive... suggest changing to he required non invasive...
Line 88: The permanence of the nasal device, for a long time... change to.. The prolonged use of CPAP...
Line 89: ...newborns have caused the failure.. suggest changing to.... newborns, caused the destruction of the baby's nasal septum.
Lines 89-90: According to the recent progress described... suggest changing to.. Recent reports described in the literature...
Line 90: There is a typo.... thr should be the
Line 91: Suggest removing the word this in front of prompted.. Also ...to try to use it in the damaged.... change to... to try to use it on the damaged
Materials and Methods:
Lines 94-95: .. with surprising results both after a few days but above all.... suggest changing to... with changes seen even after a few days and with vast improvement after 5-6 months.
Line 96: Change In according simply to According to
Results:
Line 120-121: Do the authors mean..... even if not complete regeneration.... not sure what they mean by complete restitutio ad integrum.
Line 124: Add the word the in front of orange area. Also add an s to the end of the word correspond
Lines 124-125... Add the word the in front of the word black area. Also add corresponds in front of to nostrils...It should read.... the black area corresponds to the nostrils
Line 125: I am unsure what the authors mean by.... we have used then with the function measure.... please re-word and clarify
Lines 126: We used as a reference... suggest changing to.. Included as a referance is the photo of the nasal septum....
Line 131: In graph it is represented... suggest changing entire sentence to... The reconstructed area is portrayed in red in the figure below.
Line 140: Remove the in front of Longo et al
Line 164: Remove the word our in front of reality.. it should read.. In reality...
Round 2
Reviewer 1 Report
After reviewing the revised manuscript I do not have any further comments and recommend the publication of this interesting case report.